# Therapeutic Approach to Botulinum Injections for Hemifacial Spasm, Synkinesis and Blepharospasm

**DOI:** 10.3390/toxins14050362

**Published:** 2022-05-23

**Authors:** Gilad Yahalom, Amir Janah, Gustavo Rajz, Roni Eichel

**Affiliations:** 1Department of Neurology, Shaare Zedek Medical Center, Jerusalem 9103102, Israel; ajanah89@gmail.com (A.J.); eichel@szmc.org.il (R.E.); 2The Movement Disorders Clinic, Shaare Zedek Medical Center, Jerusalem 9103102, Israel; 3The School of Medicine, The Hebrew University, Jerusalem 91905, Israel; rajzgustavo@hotmail.com; 4Department of Neurosurgery, Shaare Zedek Medical Center, Jerusalem 9103102, Israel

**Keywords:** botulinum, efficacy, side effects, hemifacial spasm, blepharospasm, approach

## Abstract

The aim of this study was to show our therapeutic outcome of botulinum injection to the facial muscles and thereby to find the best therapeutic concept which should be embraced. The decision to treat the lower eyelid with 1-point or 2-points injection was randomly taken as there is no consensus regarding this debate. Injections of the lateral end of the upper eyelid were performed more laterally to the conventional injection point, just lateral to the conjunction of the upper and lower eyelids. Twenty-three patients (12 hemifacial spasm, 6 blepharospasm, 5 post facial palsy synkinesis) were enrolled. Data were retrieved from 112 visits between 2019 and 2022. Overall, 84.9% of the treatments had moderate or marked improvement. The most common side effect was facial weakness (11.8%). Neither ptosis nor diplopia were noted. Two-points regimen in the lower eyelid was associated with a lower risk of facial weakness (*p* = 0.01), compared to 1-point regimen, with a better therapeutic outcome as reflected by more favorable PGI-C scores (*p* = 0.04). Injection of the pretarsal segment of the upper eyelid, just onto or even lateral to the conjunction of the upper and lower eyelids, lowers the risk of ptosis.

## 1. Introduction

Treatment with botulinum toxin (BT) for blepharospasm (BS) [1], hemifacial spasm (HFS) [2] and post facial paralysis synkinesis (PFPS) [3] is effective and safe. The efficacy and side effects profile are dependent on the experience and technique of the treating physician. Incobotulinumtoxin A (Xeomin^®^, INCO) differs from Onabotulinumtoxin A (Botox^®^, ONA) and abobotulinumtoxin A (Dysport^®^, ABO) as the formulation of INCO is free from complexing proteins [4]. Two approaches for injections of the orbicularis oculi muscle are being implemented: preseptal and pretarsal injections [1,5]. The pretarsal approach was found to be more effective and safer in terms of the rate of ptosis than the preseptal approach and is now probably being practiced by most centers [1,5]. For the anatomical differences between the two, we refer to the study of Cakmur et al. [5].

Ptosis is perhaps the most common and most debilitating side effect of BT injections into the facial muscles. The rate of occurrence and recurrence of ptosis is variable, ranging from 3.2% to 18% and is significantly dependent on the technique used [1,2,5,6]. Ptosis occurs due to unwanted spread of BT from the injection site, usually in the upper eyelid to the levator palpebrae muscle [1], but sometimes also following injections of the corrugators [7]. Other side effects of facial muscles injection include eye dryness [8], eye weakness [9], ocular itching [10], diplopia [10], blurred vision [1] and hematoma in the injection site [11], which seem to be secondary to either upper or lower eyelid injections. Smile asymmetry or mouth droop occur due to injections of the muscles generating the smile function, i.e., the lower eyelid or lower parts of the face [12]. The decision to treat the lower eyelid with 1-point or 2-points injection is randomly taken and there is no consensus regarding this debate. We aim to show our therapeutic outcome of BT injection into the facial muscles and thereby to find the best therapeutic concept which should be embraced.

## 2. Results

Thirty-two consecutive patients, diagnosed with either HFS or PFPS or BS, attended our clinic between the years 2019 and 2022. Nine patients were previously treated elsewhere and were excluded. Twenty-three naïve patients (seventeen females) were eventually enrolled and included in the analysis. Data were retrieved from 112 visits (mean of 3.6 ± 2.2 cycles per patient). Demographic and clinical characteristics are presented in Table 1. The majority of patients (*n* = 12) were diagnosed with HFS, followed by BS (*n* = 6) and PFPS (*n* = 5). The mean age at onset (AAO) was 46.6 ± 19.1 years. AAO of the BS group tended to be older than the PFPS group (55.4 ± 16.0 vs. 31.4 ± 12.0 years, *p* = 0.06). The age at first treatment was 53.8 ± 17.7 years. Treatments were performed using either INCO (53.6%) or ONA (46.4%).

The first treatment included a mean dose of 18.2 ± 13.2 U (13.1 ± 6.1 units for the HFS group). The mean dose increased over time. The mean time between injections was 99.3 ± 26.1 days. The overall Hemifacial Spasm 7 (HFS-7) score of the whole group was 4.5 ± 6.5 points and it decreased persistently over time (Figure 1) and so is the Clinician Global Impression of Severity (CGI-S) (Figure 2). The subjective treatment efficacy was 72.2 ± 26.5 percent. The treatment was more effective with increased doses of BT. Effect duration averaged 2.2 ± 0.9 months. The effect was longer for the BS group (*p* = 0.04).

As for the Patient Global impression of Change (PGI-C) score, marked improvement was most frequent and was scored 55 times (59.1%), followed by moderate and mild improvement (in 25.8% and 9.7%, respectively). On five occasions (5.4%), the patient reported no change. Overall, 84.9% of the treatments had moderate or marked improvement. There was no difference in PGI-C among the different disorders. However, following a head-to-head comparison between 1-point and 2-points injection regimen, PGI-C scores were more favorable in the 2-points regimen as 97.3% had marked or moderate improvement versus 75.4% in the 1-point regimen (*p* = 0.04).

Injection of the lateral-upper portion of the major zygomatic muscle was rarely performed and seemed to have no added value compared to that of its lower portion.

Analysis on HFS alone showed non-significant differences in the doses of both types of BT (19.8 ± 14.3 units for ONA vs. 18.0 ± 11.3 units for INCO, *p* = 0.90), non-significant difference in subjective treatment efficacy (71.0 ± 24.7 vs. 72.8 ± 31.7%, respectively; *p* = 0.44), and non-significant difference in the intervals between injections (105.3 ± 36.0 vs. 90.1 ± 10.2 days, respectively; *p* = 0.64).

Side effects from BT injections occurred in 25.3% of 93 visits. No significant difference in the rate of “any side effect” was noted between 1-point and 2-points approach. The most common side effect was facial weakness with a rate of 11.8%, followed by dry eye (9.7%) and hematoma at the injection site (5.4%) and lacrimation (5.4%) (Table 2). Interestingly, neither ptosis nor diplopia were noted. As for the side effect of facial weakness, most occurred following the 1-point approach, while much lower rates were noted after the 2-points approach (20.4% vs. 2.7%, respectively, *p* = 0.01) (Table 2).

## 3. Discussion

In this study, we aimed to evaluate the therapeutic effect of patients treated with BT for dystonia and hemispasm/synkinesis of the face. There was a markedly favorable therapeutic effect of BT treatment, with a fair side effects profile. Ptosis, probably the most common and debilitating side effect, being present in 3.2–18% [1,2,5,6], was not reported even in one single case. We suggest that injection of the pretarsal segment of the upper eyelid, just onto or even lateral to the conjunction of the upper and lower eyelids, can avoid infiltration of BT to the levator palpebrae muscle, without compromising the therapeutic effect. Lolekha et al. also had no ptosis, following 40 pretarsal injections [13]. In the study of Lolekha et al. the dose injected to each point on the upper eyelid was limited to 2.5 units only. We used maximal doses of 3 units per injection point. Nonetheless, our off-the-records analysis on non-naïve patients, who were treated with doses as high as 8 units ONA and 30 units of ABO per injection point (lateral and medial upper eyelid), was not related to ptosis or diplopia. Diplopia, an occasionally encountered and debilitating side effect, occurring in up to 5% [9], was not noted in our study. This is probably due to strict avoidance of injection of the far medial part of the lower eyelid, leading to diffusion of the toxin into the inferior oblique or the inferior rectus. Table 3 presents clinical outcomes found in our study and those of others. Blurred vision, a side effect which was found to occur in up to 10% in BS patients [1], was not reported in our cohort.

The rates of facial weakness (11.8%) and dry eye (9.7%) were relatively frequent, compared to the rates in the study of Sorgun et al. (3.6%, 0.3%, respectively) [6], which showed excellent side effects profile. Dry eye is a particularly frequent side effect in BS [14] and was absent in all our HFS patients. Sorgun et al. performed only one injection of 5 units in the lateral part of the lower eyelid, avoiding injection in the medial part. This methodology contradicts our statement that the 2-points injection is safer. The study of Lolekha et al. which adhered to the 2-points injection of the lower eyelid, supported our finding as relatively few side effects occurred following pretarsal injections, including hematoma (5%) and lacrimation (3.8%) [13]. Another option to minimize side effects, while we extrapolate our results and those of Sorgun et al. [6], is to inject the lateral part of the lower eyelid with higher doses and that of the medial part with lower doses. It is of note that in our analysis on naïve patients, only one patient had facial weakness after 2-point injection regimen, but this patient was also injected with 3 units in the upper part of major zygomaticus.

From all the above-mentioned, it seems that the 2-points injection technique of the lower eyelid is safe and preferable over 1-point injection, to avoid facial weakness of the injected side. The 2-points technique seems also to be more effective. It may make sense to inject the lower eyelid with no more than 3 units per injection site, and particular caution should be implemented when the medial part is injected.

The mean subjective treatment efficacy of 71.8% in HFS seen in our study is rather similar to that of Sorgun et al. [6]. Additionally, 84.9% and 97.3% of the treatments had moderate to marked response to treatment according to the PGI-C score for all and for the 2-points regimen, respectively and this outcome is comparable with others [2,5,15] (Table 3).

The main limitations of this study were its small sample and the short-lasting follow-up time. Hence, no firm conclusion should be drawn and a further larger study should be ensued in order to confirm its findings. The size of the study could be larger, but we decided to perform the analysis on our naïve patients only, since some of our patients were satisfied with their old pattern of injection from the previous clinic and preferred to have mild facial weakness as a consequence of the BT treatment.

We assume that more cycles would lead to a better therapeutic outcome and to less side effects. On the other hand, the short follow-up may omit cases in which injections of more sites are necessary, with possible secondary side effects that are not reflected in a short-term follow-up.

The assessment was incomplete due to language barriers in some patients. The effect duration time was inaccurate as in some patients it did not subside until next treatment.

The study cohort was heterogenous; hence, any conclusion should be taken with caution. The use of more than one toxin in a study is sub-optimal, based on the statement of Kent et al. [16], which found that there was no fixed-dose ratio conversion between INCO and ONA, and consistent with the product label and recommendations from regulatory agencies, the potency units of ONA are not interchangeable with other BT type A products. On the other hand, many studies used more than one type of toxin, and the conversion ratio of 1:1 between ONA and INCO is widely accepted [17,18,19]. To clarify this question, we performed another sub-analysis on HFS alone, in which no significant differences between ONA and INCO were found in the mean dose, in the mean interval between the injections and in the subjective treatment efficacy. Indeed, we did not make different decisions between both types of BT in our dosage calculation prior to each treatment.

## 4. Conclusions

This study is a proof of concept, in which we introduce a technical approach to reduce the risk of side effects to the minimum, without compromising the therapeutic effect. The extra-lateral injection of the upper eyelid prevents ptosis, and perhaps diplopia, without a deleterious impact on efficacy. The same holds true for the 2-points approach of lower eyelid injection, which reduces the risk of iatrogenic facial weakness. Furthermore, a larger study may shed light on this proof of concept.

## 5. Material and Methods

Consecutive patients who attended routinely the Movement Disorders Clinic and were diagnosed with HFS, PFPS or BS were enrolled. The diagnosis was made by a movement disorders specialist (GY). Only naïve patients were enrolled and those treated earlier at another clinic were excluded. Patients were treated in intervals of 3 months (except for the second cycle in which a booster was given, if needed, 3 weeks after first treatment). The study was conducted according to the guidelines of the Declaration of Helsinki, and approved by the Institutional Review Board of Shaare Zedek Medical Center (protocol code 0406, date of approval 21 May 2019). Informed consent was signed by each patient, prior to enrollment. Demographic data were noted, including age at injection date, AAO, gender, the side involved. Prior to each treatment, patients completed a questionnaire which included the subjective efficacy in percentage of the previous treatment at maximal effect (where 0% is no benefit and 100% is benefit to an asymptomatic state). In addition, effect duration and side effects of the previous treatment, if occurred, were also noted. Prior to each treatment, the following forms were also completed: PGI-C, CGI-S and the HFS-7. Clinical data, including the BT type, the dose and points of the injections were noted. Patients were treated with either ONA or INCO, diluted with 50 units per milliliter (mL) in a 31G-syringe. Patients received the same type of BT each treatment. The doses for each injection were chosen for each individual, based on clinical impression of the severity, ranging from 1 unit to 5 units per injection point. We followed the accepted line of 1:1 ratio between ONA and INCO in the treatment plan [17]. Possible injection points are depicted in Figure 3 and illustrated in Figure 4. The orbicularis oculi muscle included the following injection points: the pretarsal nasal (medial) and temporal (lateral) ends of the upper eyelid, the lateral point (temporal area), just beneath and lateral to the eyebrow and 2 cm lateral from the eye, the pretarsal nasal (medial) and temporal (lateral) ends of the lower eyelid, just on the edge of the orbital bone. In the 1-point regimen the lower eyelid injection was under the eye, midway between both inner and outer edges. The 2-points regimen was in both midway and lateral position, just under the outer edge. It is of note that injections in the lateral end of the upper eyelid were performed more laterally, compared to the conventional injection point, just lateral to the conjunction of the upper and lower eyelids. To minimize the risk of local hematoma of the lower eyelid, one should use the thumb and index fingers to stretch the skin in opposing horizontal directions, while injecting between the fingers the subcutaneous layer under the skin as superficially as possible. Other points of injection were the major zygomatic muscle (either the inferior or the upper-lateral portion), minor zygomatic muscle (also called the zygomatic head of the levator labii superior) and the risorius muscle. To best localize the minor and major zygomatic, as well as the risorius muscles, one should palpate the muscles and wait for muscular twitches. In cases of infrequent twitches, anatomical location is roughly estimated.

### Statistical Analysis

Demographic data were calculated using descriptive and frequency tables. Differences of continuous measures were calculated using Kruskal–Wallis and Mann–Whitney non-parametric tests and for categorical measures using chi-square tests. The *p*-value of the differences in complication rates among the disorders was defined as ≤0.01, using a correction for multiple comparisons. For other comparisons, *p*-value ≤ 0.05 was defined as statistically significant. The analysis was performed by SPSS v.28.

## Figures and Tables

**Figure 1 toxins-14-00362-f001:**
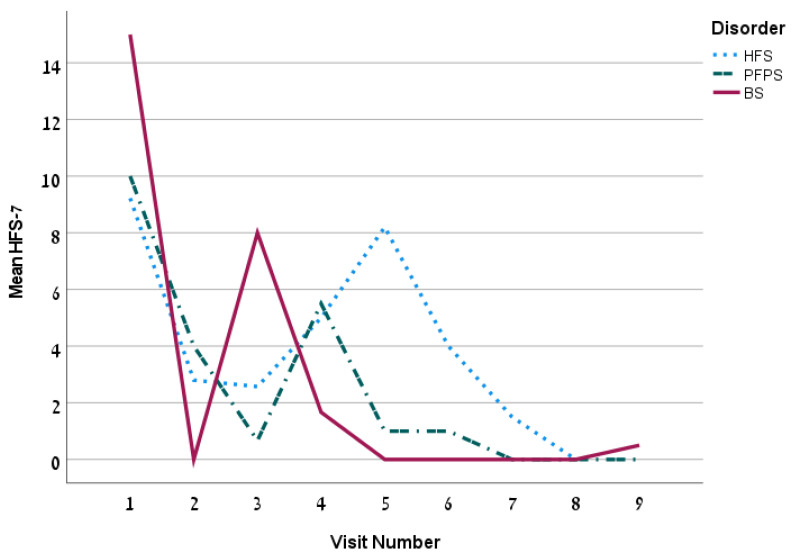
The mean HFS-7 by visit for the 3 disorders.

**Figure 2 toxins-14-00362-f002:**
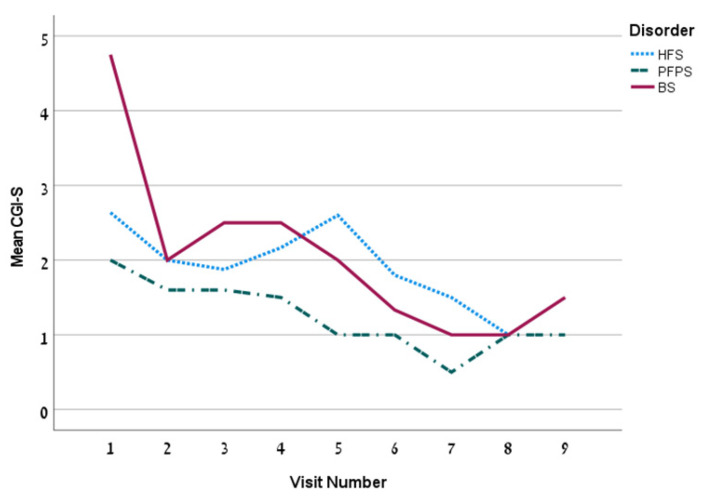
The mean CGI-S by visit for the 3 disorders.

**Figure 3 toxins-14-00362-f003:**
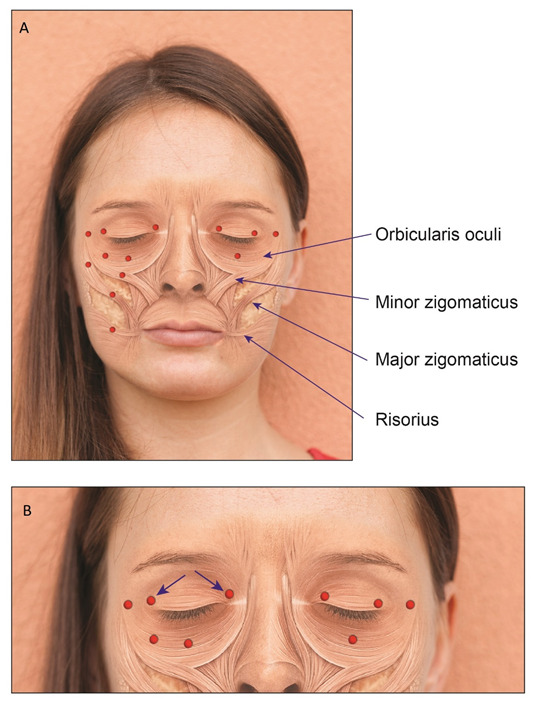
(**A**) Injection sites and anatomical locations of the relevant muscles of the upper and the lower face. The red dots indicate the injection points. Right side is our method of injection in the upper eyelid and 2-points injection of the lower eyelid. Left is the conventional method to inject the upper eyelid, with 1-point injection of the lower eyelid. (**B**) Zoom in focused on injections of the orbicularis oculi: the arrows reflect the direction of the tip of the needle. The direction of the needle for other parts is not important.

**Figure 4 toxins-14-00362-f004:**
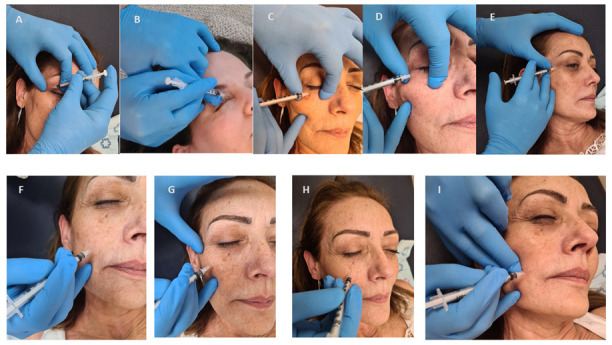
Location of injections and technique: (**A**) lateral upper eyelid (**B**) medial upper eyelid (**C**) medial lower eyelid (**D**) lateral lower eyelid (**E**) lateral part of orbicularis oculi (**F**) lower part of the major zygomaticus (**G**) upper part of the major zygomaticus (**H**) minor zygomaticus (**I**) risorius.

**Table 1 toxins-14-00362-t001:** Demographic and clinical characteristics.

			Total	HFS	PFPS	BS	*p*-Value Total
**N (% females)**			23 (73.9)	12 (58.3)	5 (100.0)	6 (83.3)	0.17
**Affected side**	Left		6	4	2	0	**<0.001**
	Right		11	8	3	0	
	Bilateral		6	0	0	6	
**Visit number**			112	49	28	35	0.99
**Time between injections, (days), mean ± SD ***			99.3 ± 26.1	99.4 ± 29.1	92.1 ± 12.4	104.3 ± 29.6	0.82
**Number of cycles, mean ± SD**			3.6 ± 2.2	3.2 ± 2.0	3.8 ± 2.4	3.9 ± 2.4	0.37
**AAO (y), mean ± SD**			46.6 ± 19.1	49.3 ± 20.0	31.4 ± 12.0	55.4 ± 16.0	0.06
**Age at first treatment (y), mean ± SD**			53.8 ± 17.7	56.6 ± 16.3	36.6 ± 12.4	62.5 ± 16.5	**0.02**
**Time from onset to treatment (y), mean ± SD**			5.6 ± 8.7	6.8 ± 11.6	5.2 ± 3.3	3.2 ± 2.3	0.64
**Total dose (U),** **mean ± SD**			27.0 ± 23.4	18.9 ± 12.8	9.0 ± 4.4	52.9 ± 22.1	**<0.001**
**Effect duration (months), mean + SD**			2.2 ± 0.9	2.1 ± 0.9	2.0 ± 0.8	2.5 ± 1.0	**0.04**
**Subjective treatment efficacy (%), mean ± SD**			72.2 ± 26.5	71.8 ± 27.6	73.5 ± 30.9	71.5 ± 21.4	0.59
**HFS-7, mean ± SD**			4.5 ± 6.5	5.3 ± 6.4	3.4 ± 5.7	4.0 ± 7.5	0.38
**CGI-S, mean ± SD**			2.0 ± 1.4	2.2 ± 1.2	1.4 ± 0.8	2.3 ± 1.8	**0.03**
**PGI-C, *n* (%)**		**Marked improvement**	55 (59.1)	20 (50.0)	16 (69.6)	19 (63.3)	0.43
		**Moderate improvement**	24 (25.8)	13 (32.5)	4 (17.4)	7 (23.3)	
		**Mild improvement**	9 (9.7)	4 (10.0)	1 (4.3)	4 (13.3)	
		**No Change**	5 (5.4)	3 (7.5)	2 (8.7)	0 (0.0)	
		**Worse**	0 (0.0)	0 (0.0)	0 (0.0)	0 (0.0)	

* Analysis was performed from the 3rd cycle. Abbreviation: N = number; AAO = age at onset; y = years; HFS = hemifacial spasm; PFPS = post facial paralysis synkinesis; BS = blepharospasm; HFS-7 = the hemifacial spasm 7 questionnaire; SD = standard deviation; U = units; CGI-S = clinician global impression of severity; PGI-C = patient’s global impression of change.

**Table 2 toxins-14-00362-t002:** Side effects by disorder for all and by the number of injection points in lower eyelid.

	**Complication’s Rate (in Percentage) by Disorder (*n* = 93)**					**Complication’s Rate (in Percentage) According to the Number of Injection Points in the Lower Eyelid**		
**Complication**	**Total**	**HFS**	**PFPS**	**BS**	***p*-Value**	**1-Point Regimen (*n* = 49)**	**2-Points Regimen (*n* = 37)**	***p*-Value**
**Facial weakness**	11.8	15.0	17.4	3.3	0.21	20.4	2.7	**0.01**
**Hematoma at injection site**	5.4	12.5	0	0	0.03	4.1	8.1	0.37
**Dry eye**	9.7	0	13.0	20.0	0.02	8.2	10.8	0.48
**Lacrimation**	5.4	5.0	8.7	3.3	0.69	2.0	8.1	0.21
**Eye size asymmetry**	1.1	2.5	0	0	0.51	2.0	0	0.57
**Ptosis**	0	0	0	0	NA	0	0	NA
**Diplopia**	0	0	0	0	NA	0	0	NA
**Any side effect per visit**	25.3	12.6	4.6	8.0	0.51	26.5	23.7	0.48
**Comparison of PGI-C scores (in percentage) between 1-point and 2-points injections of the lower eyelid**								
						**1-point regimen**	**2-points regimen**	
**Marked improvement**						54.6	68.4	**0.04**
**Moderate improvement**						20.8	28.9	
**Mild improvement**						9.3	2.6	
**No change**						8.3	0	

Abbreviation: HFS = hemifacial spasm; PFPS = post facial paralysis synkinesis; BS = blepharospasm; PGI-C = patient’s global impression of change; NA = not applicable.

**Table 3 toxins-14-00362-t003:** Clinical data from our study and from other previous studies on the therapeutic outcomes of botulinum toxin for HFS and BS.

Author	N	Diagnosis	Efficacy	Side Effects
Our study 2022	23	HFS, PFPS, BS	Subjective treatment efficacy: 72.2% Moderate to marked improvement: 84.9%	Facial weakness; 11.8% (for 2-points regimen 2.7%), hematoma: 5.4%, dry eye: 9.7%, lacrimation: 5.4% eye size asymmetry: 1.1%
Cakmur et al., 2002 [5]	53	HFS, BS	86–96% success (HFS), 90–97% (BS)	Ptosis: 7–18% (HFS), 13–16% (BS), blurred vision: 1–2% (BS), hematoma (not detailed)
Sorgun et al., 2015 [6]	68	HFS	73.7% improvement	Hematoma: 4.9%, facial asymmetry: 3.6% Ptosis: 3.6%, diplopia: 3.2%, eye weakness: 2.3%
Bentivoglio et al., 2009 [2]	108	HFS	94% success (Yes/no benefit)	Ptosis: 3.2% (ONA), 8.7% (ABO), lacrimation: 4.3% (ONA), 1.7% (ABO), irritation of conjuctivae: 2.8 (ONA), 0.6% (ABO), hematoma: 2.4% (ONA), 1.7% (ABO), blurred vision: 1.8% (ONA), 1.2% (ABO), diplopia: 1.2% (ONA), 2.3% (ABO)
Aramideh et al., 1995 [1]	45	BS	Pretarsal: 95% success (Yes/no benefit)	Diplopia: 10%, Blurred vision: 10%
Price et al., 1997 [10]	92	HFS, BS	Effect not measured	Lacrimation: 4–18%, ocular irritation: 4–18%, ptosis: 1–13%, diplopia: 1–5%
Lolekha et al., 2017 [14]	40	HFS, BS	Satisfaction rating scale: 73.8–82.8%	Ptosis: 3.8%, lacrimation: 3.8%, Hematoma: 5%, irritation: 6.3%
Poungvarin et al., 1995 [16]	55	HFS	Excellent response (80.95%) Moderate response (7.14%)	Facial weakness: 7.14%, local pain: 4.76%, lacrimation: 2.38%
Park et al., 1993 [8]	112	HFS, BS	Excellent response (98.6%)	Dry eyes: 19.8% (HFS), 27.3% (BS), mouth droop: 19.8% (HFS), ptosis: 10.9% (HFS), 27.3% (BS), lid edema: 5% (HFS), 0.9% (BS), fatigue: 4% (HFS), diplopia: 2% (HFS), 0.9% (BS), hematoma: 2% (HFS)
Cillino et al., 2010 [11]	131	HFS, BS	Effect not measured	Hematoma: 31.5% (BS), 31% (HFS), ptosis: (19.2% (BS), 17.2% (HFS), diplopia: 5.4% (BS), 8.6% (HFS), photophobia: 1.4% (BS), 3.4% (HFS), dry eyes: 2.7% (BS), 1.7% (HFS), mouth droop: 1.7% (HFS), blurred vision: 1.4% (BS)

Abbreviation: N = number; HFS = hemifacial spasm; PFPS = post facial paralysis synkinesis; BS = blepharospasm; ONA = onabotulinum toxin A; ABO = abobotulinum toxin A.

## Data Availability

The data presented in this study are available in this article.

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
