# Peer review of "Therapeutic Approach to Botulinum Injections for Hemifacial Spasm, Synkinesis and Blepharospasm"

_toxins, 2022, doi:10.3390/toxins14050362_

Round 1
Reviewer 1 Report
Comments to the authors
The topic is interesting. This study in fact showed a technique of botulinum toxin injection of patients with blepharospasm, hemispasm and facial synkinesis that by reducing adverse events manages to maintain the effectiveness of the treatment.
However, I would like to bring some observations to the authors' attention:
- In botulinum toxin treatment of patients with facial movement disorders an important barrier is adverse events. The severity of blepharospasm/facial hemispasm may be variable over time, so adding injection sites could cause adverse events in the long-term treatment of these patients. However, the results of this study are based on a few patients and for a short observation period. It would be useful to discuss this aspect in the discussion as well.
- The authors report a reduced frequency of adverse events with the addition of botulinum toxin injection sites and maintenance of efficacy. For greater clarity it would be useful to compare in a table the data of previous studies (Lolekha et al; Sorgun et al, etc.) with the results of the present study.
Author Response
Dear reviewer,
We answered the queries and have modified our paper according to your requests.
Your comments and our answers (in bold) are enclosed.
The topic is interesting. This study in fact showed a technique of botulinum toxin injection of patients with blepharospasm, hemispasm and facial synkinesis that by reducing adverse events manages to maintain the effectiveness of the treatment.
However, I would like to bring some observations to the authors' attention:
- In botulinum toxin treatment of patients with facial movement disorders an important barrier is adverse events. The severity of blepharospasm/facial hemispasm may be variable over time, so adding injection sites could cause adverse events in the long-term treatment of these patients. However, the results of this study are based on a few patients and for a short observation period. It would be useful to discuss this aspect in the discussion as well.
We noted this point in the limitation section. Page 7.
"Furthermore, the short follow-up may omit cases in which injections of more sites are necessary, with possible secondary adverse effects that are not reflected in a short-term follow-up."
- The authors report a reduced frequency of adverse events with the addition of botulinum toxin injection sites and maintenance of efficacy. For greater clarity it would be useful to compare in a table the data of previous studies (Lolekha et al; Sorgun et al, etc.) with the results of the present study.
We added Table 3 which addressed this point.
Reviewer 2 Report
Dear Authors,
This article is about a proposal of a modified approach to the treatment of hemifacial spasm, synkinesis and blepharospasm with botulinum toxin type A. Detail report on hints about major and minor revisions are reported below.
Major
- Cite more precisely the Board who approved the study protocol and if the protocol was registered.
- As reported by Kent et al., Toxins 2021, "Our results support that there is no fixed-dose ratio conversion between incobotulinumtoxinA and onabotulinumtoxinA, and consistent with the product label and recommendations from regulatory agencies, the potency units of onabotulinumtoxinA are not interchangeable with other botulinum toxin type A products" should be necessary to better described why the Authors considered interchangeable the two type of toxins used in this study.
- The Author report on the methods that the dilution used is 50U/ml. In my Country, for Botox, a precise dilution is not described in the leaflet, but it is expected for Xeomin to be 25U/ml. The leaflet in the country where the study was conducted reports a different dilution? Is the different dilution used one of the new approaches proposed?
Minor
Title
Quote in the title the design of the study.
Keywords
Prefer MeSH terms and words not already used in the title.
Introduction
The difference between the two types of BoNT-A is not described.
Material and methods
There is a possible typo on the name of the toxin "Xeomin" that is named as INO instead of INCO.
The timing of the re-injection is not clearly described.
Among the exclusion criteria is not investigated if the patients suffer from closed-angle glaucoma, which is a contraindication to use BoNT-A.
Results
Does the patient receive only toxins ONA or INCO, or between the re-injection visit the toxins were mixed?
Discussion
Why do the Authors consider them interchangeable the two types of BoNT-A are not affordable.
Table 1
The line "Dose (U+SD) injected in LEL" appears without data. The abbreviation LEL is not described anywhere in the article.
Author Response
Dear reviewer,
We answered the queries and have modified our paper according to your requests.
Your comments and our answers (in bold) are enclosed.
Dear Authors,
This article is about a proposal of a modified approach to the treatment of hemifacial spasm, synkinesis and blepharospasm with botulinum toxin type A. Detail report on hints about major and minor revisions are reported below.
Major
- Cite more precisely the Board who approved the study protocol and if the protocol was registered.
We elaborated this point in the methods section. Page 3, lines 5-8 from the heading "Material and methods".
" The study was conducted according to the guidelines of the Declaration of Helsinki, and approved by the Institutional Review Board of Shaare Zedek Medical Center (protocol code 0406, date of approval May 21, 2019). Informed consent was signed by each patient, prior to enrollment."
- As reported by Kent et al., Toxins 2021, "Our results support that there is no fixed-dose ratio conversion between incobotulinumtoxinA and onabotulinumtoxinA, and consistent with the product label and recommendations from regulatory agencies, the potency units of onabotulinumtoxinA are not interchangeable with other botulinum toxin type A products" should be necessary to better described why the Authors considered interchangeable the two type of toxins used in this study.
We added this sentence in the methods section. It is of note that we did not switch between the types of botulinum toxins. Page 3, last 4 lines.
- The Author report on the methods that the dilution used is 50U/ml. In my Country, for Botox, a precise dilution is not described in the leaflet, but it is expected for Xeomin to be 25U/ml. The leaflet in the country where the study was conducted reports a different dilution? Is the different dilution used one of the new approaches proposed?
We used the same dilution as we used for Botox, which is very comfortable since each scale line is 1 unit. This is the common practice in most if not all practitioners in my own country. Also according to the leaflet of Xeomin, this dilution is plausible (https://www.xeomin.com/healthcare-professionals/reconstitution-and-storage).
In fact, we dilute 50 units by 1.2ml saline due to loss of dead space but it is practically 50U/ml.
Minor
Title
Quote in the title the design of the study.
We did so thanks to the reviewer's comment.
Keywords
Prefer MeSH terms and words not already used in the title.
This is in this particular case unworkable to omit terms which were critical for the search, and also reflect the true.
Introduction
The difference between the two types of BoNT-A is not described.
We added a sentence in the introduction (Page 2, lines 4-6 from the heading), which describes the difference between the two.
" Incobotulinumtoxin A ((Xeomin®, INCO) differs from Onabotulinumtoxin A (Botox®, ONA) and abobotulinumtoxin A (Dysport®, ABO) as the formulation of INCO is free from complexing proteins (4)"
Material and methods
There is a possible typo on the name of the toxin "Xeomin" that is named as INO instead of INCO.
We corrected it to INCO in all relevant notes.
The timing of the re-injection is not clearly described.
We added a sentence which clarifies it in the methods section. Page 3, lines 4-5 from heading.
" Patients were treated in intervals of 3 months (except for the second cycle in which a booster was given, if needed, 3 weeks after first treatment)"
Among the exclusion criteria is not investigated if the patients suffer from closed-angle glaucoma, which is a contraindication to use BoNT-A.
Although closed-angle glaucoma was rarely described in the literature in association with BT injections and is not contraindicated in the leaflet of Xeomin and Botox (https://www.botoxmedical.com/Office/Resources), we looked through our patients and none suffered from closed-angle glaucoma. We do not think it should be noted in the text but if the reviewer thinks we have to add it, we shall do so.
Results
Does the patient receive only toxins ONA or INCO, or between the re-injection visit the toxins were mixed?
All patients in this study were given the same BT type, without switching to one another along the follow-up. We clarified it in the method section by deleting the word randomly (they were given in the first cycle randomly Botox or Xeomin). Page 3, line 4 from the end.
" Patients received the same type of BT each treatment."
Discussion
Why do the Authors consider them interchangeable the two types of BoNT-A are not affordable.
We did not consider them interchangeable and hence we treated each patient with the same type throughout the follow-up. We clarified it in the text. Page 3, line 4 from the end.
Table 1
The line "Dose (U+SD) injected in LEL" appears without data.
Indeed. We did not find the line important and we deleted it.
The abbreviation LEL is not described anywhere in the article.
We omitted it.
Reviewer 3 Report
Dear authors,
The paper demonstrated interesting research on the therapeutic outcome of botulinum injection to the facial muscles in patients with hemifacial spasm, blepharospasm and post facial paralysis synkinesis.
However, extensive revision is required.
First of all, since the article focuses on the injection points. Please clarify all the injection points that are mentioned in the text.
Secondly, prepare additional figures that indicate each injection points in the texts. Figure 1 is premature and careless.
Introduction
In line 27 please refer https://www.mdpi.com/2072-6651/14/4/268 which is recently published discussing ptosis.
Figure 1 is hard to understand, have more detailed explanation in the figure 1 legend. What does the arrows pointing out? What is the difference between left and right side of the face. Please describe.
Describe how does preseptal and pretarsal injection differs in short sentence.
What are the difference in two point injection and one point injection. Which location are they injected?
Add more figures that can describe these injection points.
Methods
I would suggest left side as conventional method, and right side as your own injection points.
I see that you have suggest major and minor zygomatic muscles with risorius muscle. How does the injection point targeted precisely for each muscles.
The injection points that has been suggested is not anatomically correct ways of targeting these muscles.
Author Response
Dear reviewer,
We answered the queries and have modified our paper according to your requests.
Your comments and our answers (in bold) are enclosed.
Dear authors,
The paper demonstrated interesting research on the therapeutic outcome of botulinum injection to the facial muscles in patients with hemifacial spasm, blepharospasm and post facial paralysis synkinesis.
However, extensive revision is required.
First of all, since the article focuses on the injection points. Please clarify all the injection points that are mentioned in the text.
Following the reviewer's comment, we changed Figure 1 and is now splitted to 1A and 1B. Figure 1A shows the anatomy. Figure 1B shows the injection points.
Secondly, prepare additional figures that indicate each injection points in the texts. Figure 1 is premature and careless.
Figure 1B addresses this point hopefiully. We tried to improve the figure. Ideally, we would put a picture from the book of Wolfgang Jost "Pictorial atlas of botulinum toxin injection" 2nd edition, 2012 (Quitessence publishing), but due to copyright issues, we preferred to do it by painting. Furthermore, we added another Figure: figure 2 which shows injection technique for each injection point.
Introduction
In line 27 please refer https://www.mdpi.com/2072-6651/14/4/268 which is recently published discussing ptosis.
We added this reference.
Figure 1 is hard to understand, have more detailed explanation in the figure 1 legend. What does the arrows pointing out? What is the difference between left and right side of the face. Please describe.
We clarified the figure legend.
Describe how does preseptal and pretarsal injection differs in short sentence.
We describe it shortly in the introduction but added the following: "For the anatomical differences between the two, we refer to the study of Cakmur et al.". Page 2, line 9-10 from the heading "introduction".
What are the difference in two point injection and one point injection. Which location are they injected?
We describe it better now, following the change in Fig.1.
Add more figures that can describe these injection points.
Following your comments, we changed figure 1 and splitted Figure 1 into 1A (anatomical orientation) and 1B (injection scheme). In addition, we added another Figure – Figure 2, to illustrate the technique.
Methods
I would suggest left side as conventional method, and right side as your own injection points.
So we did.
I see that you have suggest major and minor zygomatic muscles with risorius muscle. How does the injection point targeted precisely for each muscles.
We added a sentence in the methods section, clarifying this (Page 4, lines 17-19): "To best localize the minor and major zygomatic, as well as the risorius muscles, one should palpate the muscles and wait for muscular twitches. In cases of infrequent twitches, anatomical location is roughly estimated." Furthermore, we show in the new figures 1+2 the precise injection site.
The injection points that has been suggested is not anatomically correct ways of targeting these muscles.
All targetings were according to the book of Wolfgang Jost "Pictorial atlas of botulinum toxin injection" 2nd edition, 2012 (Quitessence publishing), with two exceptions: we injected very rarely the upper portion of the major zygomaticus, to avoid droop mouth. The common practice is to inject the lower part of the major zygomaticus but we do not see any reason not to inject its upper part, particularly if one can feel the twitches at that point. Indeed, Lee et al. injected the upper part of the major zygomaticus. Our impression was that this target was rather desperating though, with no added value compared to its lower portion.
There is little described in the literature on injections of the minor zygomaticus but in some studies it was injected at similar location as we did (Prutthipongsit et al., Lee et al.) and Jost shows the same location in his book.
Jost did not inject the lateral portion of the orbicularis oculi but many others inject this point (e.g. Poungvarin et al. 1995, Cakmur et al. 2001).
Reference:
Prutthipongsit A, Aui-aree N. The Difference of Treatment Results between Botulinum Toxin A Split Injection Sites and Botulinum Toxin A Non-Split Injection Sites for Hemifacial Spasm. J Med Assoc Thai. 2015 Nov;98(11):1119-23.
Lee JM, Choi KH, Lim BW, Kim MW, Kim J. Half-mirror biofeedback exercise in combination with three botulinum toxin A injections for long-lasting treatment of facial sequelae after facial paralysis. J Plast Reconstr Aesthet Surg. 2015 Jan;68(1):71-8.
Round 2
Reviewer 2 Report
Dear Authors,
I have read the answer both to major and minor issues.
If the latters appear almost all exhaustively overcome, it cannot be considered the same for the former. Particularly, in the study were analysed as equal both patients treated with ONA or INCO BoNT-A. In line with the latest literature on this field, "Kent et al., Toxins 2021", the two types of toxins are not interchangeable and so not comparable. A possible solution might be to split the sample and analyse the data separately.
Author Response
Dear reviewer,
Please, see the hereby response to your concerns. We have modified our paper accordingly.
Your comment and our answer (in bold) are enclosed.
Dear Authors,
I have read the answer both to major and minor issues.
If the latters appear almost all exhaustively overcome, it cannot be considered the same for the former. Particularly, in the study were analysed as equal both patients treated with ONA or INCO BoNT-A. In line with the latest literature on this field, "Kent et al., Toxins 2021", the two types of toxins are not interchangeable and so not comparable. A possible solution might be to split the sample and analyse the data separately.
To seperate the cohort to 2 cohorts with one toxin or to limit the paper to 1 toxin means that the sample of the paper which is even now not too big, will become even smaller. It is not uncommon to use 2 or more toxins in one study. There are many studies which used ABO and ONA with a conversion ratio from 1:2.5 to 1:5, usually 1:3 (Scaglione). INCO is a relatively new toxin but it is thought to have a ratio of 1:1 or 1:1.2 (Botox:Xeomin) (Scaglione, Samizadeh, Contarino). Table 2 in the paper of Scaglione presents a list of many studies which used more than 1 toxin in the same study.
We moved the sentence about Kent from the Methods section to the discussion and we discussed the point you mentioned in the limitation section. Page 7, lines 15-20.
" The use of more than one toxin in a study is sub-optimal, based on the statement of Kent et al. [17], which found that there is no fixed-dose ratio conversion between INCO and ONA, and consistent with the product label and recommendations from regulatory agencies, the potency units of ONA are not interchangeable with other BT type A products. On the other hand, many studies used more than one type of toxins, and the conversion ratio of 1:1 between ONA and INCO is widely accepted [13, 18, 19]."
Reference:
Scaglione F. Conversion Ratio between Botox®, Dysport®, and Xeomin® in Clinical Practice. Toxins (Basel) 2016 Mar 4;8(3):65.
Samizadeh S, De Boulle K. Botulinum neurotoxin formulations: overcoming the confusion. Clin Cosmet Investig Dermatol. 2018 May 30;11:273-287.
Contarino MF, Van Den Dool J, Balash Y, Bhatia K, Giladi N, Koelman JH, Lokkegaard A, Marti MJ, Postma M, Relja M, Skorvanek M, Speelman JD, Zoons E, Ferreira JJ, Vidailhet M, Albanese A, Tijssen MA. Clinical Practice: Evidence-Based Recommendations for the Treatment of Cervical Dystonia with Botulinum Toxin. Front Neurol. 2017 Feb 24;8:35.
Reviewer 3 Report
I acknowledge the effort the authors have made.
However, figure 1a and figure 2b is still a major problem.
Figure 1a and figure 2b should be in one figure.
Also, the drawing is terrible to be accepted in the "Toxins".
This is not a high school report.
Please outsource the figure to professional drawer.
There are so much ways to do this, I feel like you have showed carelessness in revising the figures.
Author Response
Dear reviewer,
We updated Figure 1 according to your important comments.
Your comments and our answer (in bold) are enclosed.
I acknowledge the effort the authors have made.
However, figure 1a and figure 2b is still a major problem.
Figure 1a and figure 2b should be in one figure.
Also, the drawing is terrible to be accepted in the "Toxins".
This is not a high school report.
Please outsource the figure to professional drawer.
There are so much ways to do this, I feel like you have showed carelessness in revising the figures.
Following your important comment, we used a professional drawer and changed Figure 1 (A+B) to a new "Figure 1" which we firmly believe to be adequate now.
Round 3
Reviewer 2 Report
Dear Authors,
I have considered the Author's point of view.
Contarino et al. 2017 stated in his paper that "LD50 tests have shown 1:1 potency ratio of incobotulinumtoxinA vs. onabotulinumtoxinA (33), and 2.3:1 of abobotulinumtoxinA vs. onabotulinumtoxinA. These data however cannot be directly translated into the clinical practice (34)." And as conclusion: "Some of the main clinical questions, including the dose equivalence between different formulations and the minimum safe treatment intervals, are matter of discussion already for several years. This knowledge gap could only be addressed by research groups willing to engage in well designed and adequately powered clinical studies".
Scaglione F. 2016 based his conclusion on the ONA and INCO ratio on evidence that came from 2005 to 2011.
As could be readable in the following articles:
- Ferrari A, Manca M, Tugnoli V, Alberto L. Pharmacological differences and clinical implications of various botulinum toxin preparations: a critical appraisal. Funct Neurol. 2018 Jan/Mar;33(1):7-18. doi: 10.11138/fneur/2018.33.1.007. PMID: 29633692; PMCID: PMC5901944.
- Wilson AJ, Chang B, Taglienti AJ, Chin BC, Chang CS, Folsom N, Percec I. A Quantitative Analysis of OnabotulinumtoxinA, AbobotulinumtoxinA, and IncobotulinumtoxinA: A Randomized, Double-Blind, Prospective Clinical Trial of Comparative Dynamic Strain Reduction. Plast Reconstr Surg. 2016 May;137(5):1424-1433. doi: 10.1097/PRS.0000000000002076. PMID: 27119918.
- Thomas AJ, Larson MO, Braden S, Cannon RB, Ward PD. Effect of 3 Commercially Available Botulinum Toxin Neuromodulators on Facial Synkinesis: A Randomized Clinical Trial. JAMA Facial Plast Surg. 2018 Mar 1;20(2):141-147. doi: 10.1001/jamafacial.2017.1393. PMID: 28973094; PMCID: PMC5885958.
- Rupp D, Nicholson G, Canty D, Wang J, Rhéaume C, Le L, Steward LE, Washburn M, Jacky BP, Broide RS, Philipp-Dormston WG, Brin MF, Brideau-Andersen A. OnabotulinumtoxinA Displays Greater Biological Activity Compared to IncobotulinumtoxinA, Demonstrating Non-Interchangeability in Both In Vitro and In Vivo Assays. Toxins (Basel). 2020 Jun 13;12(6):393. doi: 10.3390/toxins12060393. PMID: 32545832; PMCID: PMC7354455.
- Kent R, Robertson A, Quiñones Aguilar S, Tzoulis C, Maltman J. Real-World Dosing of OnabotulinumtoxinA and IncobotulinumtoxinA for Cervical Dystonia and Blepharospasm: Results from TRUDOSE and TRUDOSE II. Toxins (Basel). 2021 Jul 14;13(7):488. doi: 10.3390/toxins13070488. PMID: 34357959; PMCID: PMC8310174.
The two tipe of BoNT-A are not interchangeble. Furthermore, this evidence is already known when the articles cited by the Authors were published and is confirmed in more recent articles. This confirms that, although the two types of toxins can be safely interchanged with each other, to achieve the same dose-related effects the patients could undergo a correction of the dose to achieve the same effects and the same duration. Indeed, besides the dose ratio, also the lasting effect of an equal dose of INCO and ONA is different (INCO<ONA).
Based on this more recent evidence it might be useful to provide in the paper the partial data (mean dose of each type of BoNT-A used, the mean time of the reinjection; means score of the efficacy outcome measures used). If the differences between the two groups will not significant, clearly expressing the limits of the study in the discussion, the data might be also considered together. If a significant difference will come out the data should be considered only as separate groups.
Author Response
Dear reviewer,
Please, see the hereby response to your concerns. We have modified our paper accordingly.
Your comment and our answer (in bold) are enclosed.
Dear Authors,
I have considered the Author's point of view.
Contarino et al. 2017 stated in his paper that "LD50 tests have shown 1:1 potency ratio of incobotulinumtoxinA vs. onabotulinumtoxinA (33), and 2.3:1 of abobotulinumtoxinA vs. onabotulinumtoxinA. These data however cannot be directly translated into the clinical practice (34)." And as conclusion: "Some of the main clinical questions, including the dose equivalence between different formulations and the minimum safe treatment intervals, are matter of discussion already for several years. This knowledge gap could only be addressed by research groups willing to engage in well designed and adequately powered clinical studies".
Scaglione F. 2016 based his conclusion on the ONA and INCO ratio on evidence that came from 2005 to 2011.
As could be readable in the following articles:
- Ferrari A, Manca M, Tugnoli V, Alberto L. Pharmacological differences and clinical implications of various botulinum toxin preparations: a critical appraisal. Funct Neurol. 2018 Jan/Mar;33(1):7-18. doi: 10.11138/fneur/2018.33.1.007. PMID: 29633692; PMCID: PMC5901944.
- Wilson AJ, Chang B, Taglienti AJ, Chin BC, Chang CS, Folsom N, Percec I. A Quantitative Analysis of OnabotulinumtoxinA, AbobotulinumtoxinA, and IncobotulinumtoxinA: A Randomized, Double-Blind, Prospective Clinical Trial of Comparative Dynamic Strain Reduction. Plast Reconstr Surg. 2016 May;137(5):1424-1433. doi: 10.1097/PRS.0000000000002076. PMID: 27119918.
- Thomas AJ, Larson MO, Braden S, Cannon RB, Ward PD. Effect of 3 Commercially Available Botulinum Toxin Neuromodulators on Facial Synkinesis: A Randomized Clinical Trial. JAMA Facial Plast Surg. 2018 Mar 1;20(2):141-147. doi: 10.1001/jamafacial.2017.1393. PMID: 28973094; PMCID: PMC5885958.
- Rupp D, Nicholson G, Canty D, Wang J, Rhéaume C, Le L, Steward LE, Washburn M, Jacky BP, Broide RS, Philipp-Dormston WG, Brin MF, Brideau-Andersen A. OnabotulinumtoxinA Displays Greater Biological Activity Compared to IncobotulinumtoxinA, Demonstrating Non-Interchangeability in Both In Vitro and In Vivo Assays. Toxins (Basel). 2020 Jun 13;12(6):393. doi: 10.3390/toxins12060393. PMID: 32545832; PMCID: PMC7354455.
- Kent R, Robertson A, Quiñones Aguilar S, Tzoulis C, Maltman J. Real-World Dosing of OnabotulinumtoxinA and IncobotulinumtoxinA for Cervical Dystonia and Blepharospasm: Results from TRUDOSE and TRUDOSE II. Toxins (Basel). 2021 Jul 14;13(7):488. doi: 10.3390/toxins13070488. PMID: 34357959; PMCID: PMC8310174.
The two tipe of BoNT-A are not interchangeble. Furthermore, this evidence is already known when the articles cited by the Authors were published and is confirmed in more recent articles. This confirms that, although the two types of toxins can be safely interchanged with each other, to achieve the same dose-related effects the patients could undergo a correction of the dose to achieve the same effects and the same duration. Indeed, besides the dose ratio, also the lasting effect of an equal dose of INCO and ONA is different (INCO<ONA).
Based on this more recent evidence it might be useful to provide in the paper the partial data (mean dose of each type of BoNT-A used, the mean time of the reinjection; means score of the efficacy outcome measures used). If the differences between the two groups will not significant, clearly expressing the limits of the study in the discussion, the data might be also considered together. If a significant difference will come out the data should be considered only as separate groups.
We performed additional analyses comparing the outcomes of ONA vs. INCO. As we noted in the limitations in the very beginning, the study is heterogenous and small and should be regarded as a proof of concept.
Initially, we did analysis comparing the doses, intervals between injections, and subjective treatment efficacy between ONA and INCO. Doses were much higher for the ONA group, but we noted that ONA was injected in a significantly higher proportion to BS while INCO was injected more frequently to PFPS and for HFS both types were equally administered (see table 1).
As a result, we performed analysis on HFS alone, comparing ONA and INCO under more homogenous conditions. You can see in Table 2 that there are actually no differences in all parameters.
We added the results of the analysis on HFS alone to the "results" section. (Page 5, line 8-12 from the end in the "Results").
"In addition, analysis on HFS alone showed non-significant differences in the doses of both types of BT (19.8±14.3 units for ONA vs. 18.0±11.3 units for INCO, p=0.90), non-significant difference in subjective treatment efficacy (71.0±24.7 vs. 72.8±31.7%, respectively; p=0.44), and non-significant difference in the intervals between injections (105.3±36.0 vs. 90.1±10.2 days, respectively; p=0.64)."
We further added a sentence clarifying this point to the discussion in the limitations section. (Page 7, 2nd line from the end to Page 8, lines 1-3).
" To clarify this question, we performed another sub-analysis on HFS alone, in which no significant differences between ONA and INCO were found in the total mean dose, in the mean interval between the injections and in the subjective treatment efficacy. Indeed, we did not make different decisions between both types of BT in our dosage calculation prior to each treatment."
We think that there is no need to add the tables to the manuscript, but if you think that the table(s) should be added, we shall do so.
Supplementary Table 1: Differences in clinical manifestations after injections with ONA vs. INCO for HFS, PFPS and BS.
|
|
ONA (n=52) |
INCO (n=60) |
p-value |
|
|
Time (days) between injections (mean±SD)* |
103.2±32.2 |
92.8±13.6 |
0.71 |
|
|
Total dose (units), mean±SD |
35.4±25.7 |
19.8±18.5 |
<0.001 |
|
|
Subjective treatment efficacy (%) |
71.1±21.6 |
73.3±30.8 |
0.15 |
|
|
Botulinum type by disorder |
HFS, n |
25 |
24 |
<0.001 |
|
PFPS, n |
4 |
24 |
||
|
BS, n |
23 |
12 |
||
|
SE of facial weakness, % |
Both techniques |
20.5 |
6.5 |
0.06 |
Abbreviation: ONA = onabotulinum toxin A; INCO = incobotulinum toxin A; N = number; NA = not applicable
- Analysis was performed from 3rd cycle
Supplementary Table 2: Differences in clinical manifestations after injections with ONA vs. INCO for HFS alone.
|
|
ONA (n=25) |
INCO (n=24) |
p-value |
|
|
Time (days) between injections (mean±SD)* |
105.3±36.0 |
90.1±10.2 |
0.64 |
|
|
Total dose (units), mean±SD |
19.8±14.3 |
18.0±11.3 |
0.90 |
|
|
Subjective treatment efficacy (%) |
71.0±24.7 |
72.8±31.7 |
0.44 |
|
|
SE of facial weakness, % |
Both techniques |
33.3 |
12.5 |
0.31 |
Reviewer 3 Report
I would accept their revision.
Author Response
We thank the reviewer for his approval.